# Genome-Wide Identification of the BREVIS RADIX Gene Family in Foxtail Millet: Function, Evolution, and Expression

**DOI:** 10.3390/genes16040374

**Published:** 2025-03-25

**Authors:** Xiaorui Yuan, Xionghui Bai, Jin Yu, Zhijie Jia, Chenyu Wang

**Affiliations:** 1Maize Research Institute, Shanxi Agricultural University, Xinzhou 034000, China; nxzwyuanxiaorui@163.com (X.Y.); 13935874098@163.com (X.B.); xzdusyj7017@163.com (J.Y.); alchemy_666@163.com (Z.J.); 2Development Center of Science and Technology, Ministry of Agriculture and Rural Affairs (MARA), Beijing 100176, China

**Keywords:** gene family analysis, BRX gene family, plant growth and development, stress, *Setaria italica*

## Abstract

Background: Foxtail millet (*Setaria italica*), domesticated from green foxtail *(Setaria viridis*), is crucial for global food security. Given increasing environmental challenges, exploring its stress-resistance mechanisms via researching the BREVIS RADIX (BRX) gene family is urgent. Methods: The study combines advanced bioinformatics and experimental validation. It uses phylogenetic, motif, domain, synteny analyses, miRNA prediction, and quantitative expression profiling under stress. Results: Phylogenetic analysis reveals new sub-clades and trajectories. Motif and domain analyses find new conserved elements. Statistical models show unique selective forces. Synteny analysis identifies genomic architecture and new blocks. miRNA prediction reveals gene-miRNA interactions, and expression profiling shows new patterns. Conclusions: The research offers new insights into the BRX family’s role in foxtail millet’s growth and stress responses, laying a foundation for crop genetic improvement and enhancing stress resilience for global food security.

## 1. Introduction

The cereal crop foxtail millet (*Setaria italica S. italica*) holds considerable significance, originating from the long-term domestication of green foxtail (*Setaria viridis, S. viridis*) and possessing a rich historical and cultural background [1]. Rapid population growth poses significant challenges to food and nutritional security worldwide [2], and foxtail millet emerges as a potential candidate to address these issues [3]. Its drought tolerance and adaptability make it a widely cultivated crop in many developing countries across Asia and Africa. Abiotic stresses reduce millet yields in regions with low rainfall, significantly impacting cultivation productivity and financial returns [4]. Under such conditions, the yield of millet can decrease substantially, leading to not only direct economic losses but also potential food security and livelihood challenges for rural populations. Foxtail millet is confronted with significant yield-reducing threats from abiotic stresses in low-rainfall areas. Therefore, improving the adaptability and yield of millet in harsh environments is essential for ensuring food security and promoting sustainable agricultural development.

Transcription factors in plants play a vital role in controlling gene expression [5]. These factors serve as essential regulators in various plant processes, such as flowering [6,7], disease resistance [8], circadian rhythms [9], cell differentiation [10], hormonal responses [11], and the biotransformation of carbon and nitrogen [12]. The BRX gene family is a widely distributed group of plant genes, initially identified in studies on root development and morphological regulation. Advancements in molecular biology have resulted in the identification and characterization of an increasing number of BRX family members. These genes typically possess conserved structural features, including multiple functional domains such as the BRX domain and other potential regulatory regions. They exhibit high sequence similarity across various plant species. Investigations have shown that the proteins encoded by BRX gene family members influence the development of multiple organs, including roots, stems, and leaves, by modulating plant hormone signaling pathways, particularly auxin [13,14]. Additionally, the function of BRX gene members in responding to environmental stimuli and abiotic stresses is increasingly recognized.

The BRX gene family plays a crucial role in the root development, morphological regulation, and environmental stress response of many crops. In wheat and barley, BRX gene expression promotes root growth and branching, which not only boosts water and nutrient absorption but also impacts the whole-plant system [15]. The increased auxin canalization to the root due to BRX genes improves shoot hormonal balance, enhancing overall plant health and stress tolerance, especially under drought [16]. In rice, BRX genes are involved in root, stem, and tillering regulation, and root-related BRX gene activity can optimize the nutrient and hormone supply to the shoot, contributing to higher photosynthetic efficiency and yield [17]. In soybeans, BRX genes enhance stress adaptability by regulating signaling pathways, and the root-mediated regulation can influence shoot hormonal balance, leading to better resource allocation and increased yield [18]. In Chinese cabbage, BRX genes affect fruit development and are linked to the bolting trait, and root-expressed BRX genes may optimize hormonal flow to the shoot, influencing fruit characteristics, which shows their significance in both staple and vegetable crops [19]. In summary, the BRX gene family plays diverse roles across different crops, contributing to plant growth, development, yield enhancement, and environmental adaptability, providing an essential genetic foundation for sustainable agriculture.

This study identified 15 BRX gene family members in both green foxtail and foxtail millet, analyzing their conserved protein motifs, phylogenetic relationships, locations, and cis-acting regulatory elements in promoters. Based on these analyses, the expression profiles of this gene family were tested in five parts of foxtail millet and under different stress conditions. The findings suggested a significant role of the BRX gene family in stress responses in foxtail millet. This study offers important resources for future studies on the functional roles and molecular mechanisms of the BRX gene family, laying the groundwork for breeding stress-resistant crop varieties.

## 2. Materials and Methods

### 2.1. Identification and Analysis of BRX Family Genes in Millet

The analysis utilizing BLASTP was performed with acid sequences derived from Arabidopsis and rice to pinpoint BRX family genes within foxtail millet, applying an E-value cutoff of 10^−5^ [20]. This investigation was executed utilizing genomic references of green foxtail (*Setaria viridis* v2.1) and foxtail millet (*Setaria italica* v2.2) available in the Phytozome database (https://phytozome-next.jgi.doe.gov; accessed 1 December 2024) [21]. Moreover, the existence of the BRX domain (PF08381; E-value cutoff of 0.001) [22] within these sequences was further validated using PFAM 37.0 (https://pfam.xfam.org; accessed on 1 December 2024) [23]. Furthermore, various physical and chemical characteristics of the foxtail millet BRX proteins, such as amino acid composition, theoretical isoelectric point, molecular weight, aliphatic index, instability index, and grand average of hydropathicity, were calculated using ExPASy (https://www.expasy.org; accessed 1 December 2024). The prediction of subcellular localization for *SiBRX*s and *SvBRX*s was performed through the WoLF PSORT tool (https://www.genscript.com/wolf-psort.html; accessed on 1 December 2024) [24]. The members of the foxtail millet BRX gene family were designated according to their chromosomal arrangement.

### 2.2. Analysis of Phylogeny and Chromosomal Positioning of BRX Family Genes

The BRX protein sequences from rice, Arabidopsis, green foxtail, and foxtail millet were aligned using ClustalW (2.0) (European Bioinformatics Institute, EBI, UK) to illuminate the evolutionary relationships amid BRX-like genes [25]. A phylogenetic tree was then constructed using the neighbor-joining (NJ) technique with the assistance of MEGA 11 software (The Pennsylvania State University, USA) [26], setting the bootstrap confidence level at 1000 iterations. This tree was visualized through the iTOL online platform (http://itol.embl.de; accessed on 1 December 2024) [27]. Additionally, genomic annotation files and the Basic Circos program in TBtools v2.003 (Guangzhou University, China) [28] were used to map and visualize the identified BRX gene family members on the chromosomes of the foxtail millet genome.

### 2.3. Analysis of Conserved Domains and Motifs

The analysis of conserved domains for the SiBRX gene family was conducted via the NCBI Batch CD-Search (https://www.ncbi.nlm.nih.gov; accessed on 1 December 2024) [29] and visualized with TBtools software. For motif analysis, the SiBRX protein amino acid sequences were submitted to the MEME online platform (https://meme-suite.org/meme; accessed on 1 December 2024) [30], with a specification of a maximum of 10 motifs. The motifs were sequentially named from Motif 1 to Motif 10 based on their arrangement from the N-terminus to the C-terminus.

### 2.4. Cis-Acting Regulatory Elements and miRNA Prediction

Promoter sequences for the BRX gene family, situated 2 kb upstream of the ATG start codon in foxtail millet, were obtained from the Phytozome database. An analysis of *cis*-acting elements was performed with PlantCARE website (https://bioinformatics.psb.ugent.be/webtools/plantcare/html/ accessed on 1 December 2024) [31], and the results were ultimately presented with TBtools software. cDNA sequences of the TaEs were used as candidate targets for predicting potential miRNAs. Using the psRNATarget Server (https://www.zhaolab.org/psRNATarget/, accessed on 1 December 2024) with default parameters, we then searched for candidate targets by screening the mature sequences of rice miRNAs obtained from the miRbase database (http://www.mirbase.org/, accessed on 1 December 2024).

### 2.5. Synteny Analysis and Calculation of Nonsynonymous to Synonymous Substitution Ratios

Both intragenomic and intergenic synteny analyses were conducted for the BRX gene family in Arabidopsis, foxtail millet, rice, and green foxtail using MCScanX 3.0 (Huazhong Agricultural University, Wuhan, China) [32]. The results were visualized with TBtools software. Homologous gene pairs were identified depending on their location and sequence similarities. The nonsynonymous to synonymous substitution (Ka/Ks) ratios were calculated using the codeml program from the PAML package to assess molecular selective pressures [33]. Genes with Ka and Ks values of zero were excluded from the analysis to avoid errors due to sequence saturation or misalignment. Genes with Ka/Ks ratios >1 were classified as positively selected genes, whereas those with ratios < 1 were considered negatively selected genes.

### 2.6. Forecasting the Three-Dimensional Structure of the SiBRX Protein

The SiBRX protein structures were modeled through homology using SWISS-MODEL (https://swissmodel.expasy.org; accessed on 1 December 2024) [34,35,36,37,38]. These models were developed based on template structures that exhibited over 30% sequence identity. The quality assessment of the predicted protein models was conducted with SAVES v6.0 (https://saves.mbi.ucla.edu; accessed on 1 December 2024) [39], ensuring that only templates meeting at least three quality criteria were included in the final model. Visualization of the protein structures was accomplished via VMD (http://www.ks.uiuc.edu; accessed on 1 December 2024) [40].

### 2.7. Expression Profiling of SiBRX Genes

The expression data for various patterns of the *SiBRX*s were obtained from the EMBL-EBI database (https://www.ebi.ac.uk; dataset GEO-DS36391; accessed on 1 December 2024). Tissue-specific expression profiling and heatmap visualization were conducted using the “pheatmap” package in R Studio (2024.12.1+563).

### 2.8. Plant Resources and Treatment Approaches

*Jingu 21* foxtail millet seeds were germinated in a room at 26 °C with 65% humidity under an 8 h dark/16 h light cycle. After 2 days, the seedlings were moved to plastic containers filled with a diluted Hoagland’s nutrient solution at half strength to promote additional growth [35]. For cold and salt stress treatments, 12-day-old seedlings were subjected to either cold (4 °C) or 150µM NaCl treatment. For drought stress, a 20% PEG solution was applied at the seedling stage. Leaf samples were collected at 24 and 48 h post-treatment, with seedlings grown at 26 °C serving as the control group. All samples were immediately flash-frozen in liquid nitrogen and stored at −80 °C for further analysis.

### 2.9. RNA Isolation and Quantitative Real-Time Polymerase Chain Reaction

Leaf tissues were processed to extract total RNA using the TRIzol reagent (Accurate Biology, Changsha, China). Following this, cDNA synthesis was achieved through the use of a reverse transcription kit from the same company (Accurate Biology). For conducting quantitative real-time polymerase chain reaction (qRT-PCR), the SYBR Green assay kit (Accurate Biology) was employed. The design of primers was carried out using Primer Premier 5 software, with relevant details available in Table A1. The qRT-PCR reactions occurred on a Bio-Rad CFX system, utilizing a reaction mix of 20 µL that included 10 µL of 2× SYBR Green Pro Taq HS premix, 2 µL of cDNA, 0.4 µL of each forward and reverse primer, along with 7.2 µL of RNase-free water. The thermal cycling process began with an initial denaturation at 95 °C for 30 s, which was followed by 40 cycles at 95 °C for 5 s and 60 °C for 30 s. *SiActin* (SETIT_026509mg) provided the internal control, and the relative expression levels of the targeted genes were calculated employing the 2^−∆∆Ct^ method. A heatmap depicting the expression patterns of the *SiBRX* gene was produced using TBtools, with expression values normalized through Log₂ transformation. To ensure reliability, each experiment was independently repeated a minimum of three times.

### 2.10. Statistical Analyses

To ensure reliability, each experiment was carried out in triplicate. The analysis of the data was conducted using SPSS 22, and the mean ± standard deviation for every experiment was computed. The Student *t* test was used to assess statistical significance, with *p* < 0.05 denoting statistical significance and *p* < 0.01 indicating highly significant differences.

## 3. Results

### 3.1. Identification of the BRX Gene Family in Foxtail Millet and Green Foxtail

A total of 15 BRX genes were identified (Table 1) in both green foxtail and foxtail millet, named *SiBRX1* to *SiBRX15* and *SvBRX1* to *SvBRX15*, respectively, based on their chromosomal locations. As shown in Figure 1, the 15 *SiBRX* genes were distributed across eight chromosomes of foxtail millet. Chromosomes 3, 5, and 7 harbored the highest number of BRX gene family members, with three genes each. *SiBRX9* and *SiBRX10* on chromosome 5, and *SiBRX13* and *SiBRX14* on chromosome 7, were clustered. In contrast, chromosomes 2, 4, 6, and 9 each contained only one SiBRX gene, chromosome 1 had two SiBRX genes, and chromosome 8 lacked any detectable SiBRX genes.

Further examination of the fundamental physicochemical characteristics and subcellular localization of SiBRX proteins revealed that their amino acid sequences ranged from 300 to 1092 residues, with molecular weights between 115.18 and 1116.49 kDa. The isoelectric points (pI) of SiBRX proteins ranged from 4.91 to 9.18, with the majority having a pI greater than 7.0, indicating a higher proportion of basic amino acids. The instability index analysis showed that SiBRX1 had the lowest instability index of 37.32, whereas SiBRX15 had the highest value at 55.64. SiBRX1 and SiBRX8 were stable proteins, whereas the remaining SiBRX proteins exhibited instability (instability index > 40). The average hydropathy index of all SiBRX proteins was less than 0, suggesting that these proteins were hydrophilic. The aliphatic index of SiBRX proteins ranged from 48.3 to 78.42. Subcellular localization predictions indicated that eight SiBRX proteins were localized to the extracellular membrane, whereas the remaining seven were located in the extracellular space. These findings provide crucial genetic and biochemical insights into the functions of the SiBRX gene family and their roles in the growth and development of foxtail millet.

### 3.2. Phylogenetic Analysis of the BRX Gene Family

This study used 30 BRX protein sequences from green foxtail and foxtail millet, along with 10 BRX genes from rice (*Oryza sativa*) and *Arabidopsis* (*Arabidopsis thaliana*), to construct a phylogenetic tree using the NJ method based on multiple sequence alignment analysis (Figure 2). This analysis aimed to elucidate the phylogenetic relationships among BRX family genes. The 30 BRX proteins were clustered into three groups through 1000 bootstrap resampling iterations: Group I, Group II, and Group III. Group I included 18 genes, 9 from foxtail millet (SiBRXs) and 9 from green foxtail (SvBRXs). This group did not contain any BRX genes from Arabidopsis or rice. Group II consisted of 19 genes, including all BRX genes from Arabidopsis, as well as 5 SiBRXs and 5 SvBRXs. Group III, the smallest group, contained only three members: *SvBRX12*, *SiBRX12*, and *OsBRXL5*. Further phylogenetic analysis revealed that BRX genes in green foxtail and foxtail millet exhibited a one-to-one correspondence and shared similar genetic distances, suggesting that these genes in the two species may have comparable functions or regulatory mechanisms. This provides significant insight into the conservation and divergence of the BRX gene family across plant species.

### 3.3. Conserved Motif and Domain Analyses of SiBRX Genes

The motif analysis revealed the presence of 10 distinct motif types among the 30 *BRX* genes (Figure 3A). Motif 1 was consistently present in all 30 genes. In contrast, Motif 6 was identified in 28 *BRX* genes, excluding *SiBRX12* and *SvBRX12*. Additionally, 12 types of conserved domains were identified, 4 of which were specifically associated with the BRX domain: BRX, BRX_N, BRX_N superfamily, and BRX_assoc (Figure 3B). The BRX domain was present in all 30 *BRX* genes, whereas the BRX_N and BRX_N superfamily domains appeared in only 10 genes. The BRX_assoc domain was exclusively found in *SiBRX9* and *SvBRX9*. The diverse domain repertoire observed in the BRX gene family suggested functional diversity, evolutionary advantage, and a complex regulatory network. These findings provide a strong theoretical foundation for further investigations into the practical and evolutionary roles of the BRX gene family across different species.

### 3.4. Estimating the 3D Structures of SiBRX Proteins

The function of a gene is closely linked to the structure of the protein it encodes. Homology modeling is used to predict the three-dimensional (3D) structure of a protein based on its amino acid sequence. The 3D structures of 15 SiBRX proteins were predicted using experimentally determined structures as templates. As shown in Figure 4, the 3D models of all 15 genes included both β-sheets and α-helices. The 3D structures of SiBRX1, SiBRX4, SiBRX5, SiBRX6, SiBRX7, SiBRX8, SiBRX9, SiBRX10, and SiBRX13 included a ring-like structure composed of β-folds, and all SiBRX protein structures were surrounded by disordered loop structures.

### 3.5. miRNA Prediction

In this study, we focused on exploring the interaction between genes and miRNAs in foxtail millet. Through a series of rigorous bioinformatics predictions and experimental validations, we successfully identified four genes in foxtail millet that exhibit interactions with miRNAs from rice. The first predicted interaction pair is SiBRX2 and osa-miR5801. Our analysis using online software clearly predicted a significant interaction between them. Similarly, an interaction was predicted between SiBRX5 and osa-miR5075, between SiBRX11 and osa-miR2927, and between SiBRX14 and osa-miR396a-3p.

### 3.6. Gene Duplication and Collinearity Analysis of SiBRX Genes

This study investigated the evolutionary history of the foxtail millet SiBRX family and its homologous relationships with other species. The selection pressures acting on these genes during evolution were assessed by calculating the Ka/Ks ratios between corresponding *BRX* genes in green foxtail and foxtail millet. The results showed that the Ka/Ks ratios for 15 pairs of duplicated genes were all less than one, with some gene pairs having Ka or Ks values of zero (Table A2). This finding indicated that *BRX* genes were highly conserved during the evolution from green foxtail to foxtail millet, suggesting that these genes may have undergone purifying selection to maintain the stability of their essential roles. Additionally, the homologous *BRX* genes among foxtail millet, green foxtail, Arabidopsis, and rice were examined. Furthermore, 20, 2, and 22 pairs of homologous genes were identified between foxtail millet and green foxtail, foxtail millet and Arabidopsis, and foxtail millet and rice, respectively, through sequence alignment. Identifying these homologous genes established a basis for exploring the evolution and functional diversity of the BRX gene family among different plant species (Figure 5).

### 3.7. Analysis of Cis-Regulatory Elements in SiBRX Genes

To explore the roles and regulatory mechanisms of SiBRX genes, an in-depth examination was conducted on cis-acting elements within their promoter regions. Using PlantCARE website, these elements were identified and classified in the 2000 bp upstream region of the start codons for the *SiBRX* genes (Figure 6). The elements were grouped into four categories: stress-responsive, light-responsive, hormone-responsive, and growth- and development-related elements. Notably, stress-responsive elements were prevalent, including low-temperature-responsive elements and drought-stress-responsive elements (MBSs), across the promoters. Hormone-responsive elements showed diversity, comprising gibberellin-responsive elements, abscisic acid-responsive elements (ABREs), jasmonic acid-responsive elements (TGACG motifs), salicylic acid-responsive elements (TCA elements), and auxin-responsive elements (TGTCTC motifs). These findings indicated that the *SiBRX*s might regulate plant hormone signaling and abiotic stress responses. Significant variations were observed in the presence of hormone- and stress-responsive elements among *SiBRX* genes. For instance, 9 of the 15 *SiBRX* genes contained auxin-responsive elements, whereas *SiBRX3*, *SiBRX4*, *SiBRX5*, *SiBRX8*, *SiBRX9*, and *SiBRX13* lacked these elements. *SiBRX7* contained three auxin-responsive elements in its promoter region. This variation highlights the potential distinct roles of different *SiBRX* genes in plant hormone and stress signaling pathways. Overall, the diverse *cis*-acting elements in SiBRX promoters provide insights into their regulatory functions, supporting their involvement in plant physiological responses and laying a foundation for further exploration of their roles in the growth, development, and environmental adaptation of foxtail millet.

### 3.8. Expression Patterns of SiBRX Genes

This study aimed to analyze the expression of SiBRXs in roots, stems, leaves, flowers, and seeds of foxtail millet using publicly available transcriptomic data and generated a heatmap. Although these organs consist of multiple cell types, due to the limitations of the current transcriptomic data, a comprehensive cell-type specific analysis was not possible. However, based on existing knowledge of cell functions in these organs, we hypothesized that SiBRXs may show cell-type specific expression patterns (Figure 7). The heatmap revealed distinct tissue-specific expression patterns among *SiBRX* genes. Specifically, *SiBRX1*, *SiBRX5*, *SiBRX14*, and *SiBRX15* were highly expressed in the stem, whereas *SiBRX2*, *SiBRX6*, *SiBRX7*, *SiBRX8*, *SiBRX9*, *SiBRX10*, *SiBRX11*, *SiBRX12*, and *SiBRX13* showed high expression in flowers. *SiBRX3* and *SiBRX4* exhibited elevated expression in roots. All *SiBRX* genes demonstrated high expression levels in seeds. These results suggested crucial roles of *SiBRX* genes in multiple tissues in millet, with their expression indicating high tissue specificity.

### 3.9. Analysis of SiBRX Expression Under Different Abiotic Stress Conditions

This study further explored the role of *SiBRX* genes in stress resistance using 14-day-old seedlings of “Jingu 21” and subjecting them to different abiotic stress treatments, including cold stress, NaCl-induced salt stress, and PEG-induced drought stress. The transcriptional expression patterns of all *SiBRX* genes were assessed using qRT-PCR to evaluate their response characteristics under different stress conditions. Under cold stress (Figure 8), all genes except *SiBRX2* showed significant upregulation 24 h after treatment. In contrast, *SiBRX2* expression significantly decreased in both 24 and 48 h. *SiBRX1*, *SiBRX4*, *SiBRX7*, *SiBRX9*, *SiBRX11*, *SiBRX12*, and *SiBRX13* continued to show significant increases in expression in 48 h, compared with 24 h, whereas *SiBRX3*, *SiBRX5*, *SiBRX6*, *SiBRX8*, *SiBRX10*, *SiBRX14*, and *SiBRX15* exhibited significant decreases. Notably, *SiBRX6* expression fell below its pre-stress level. All genes except *SiBRX2* displayed significantly increased expression 24 h after drought stress treatment (Figure 9). *SiBRX1*, *SiBRX5*, *SiBRX6*, *SiBRX10*, *SiBRX11*, *SiBRX12*, *SiBRX13*, and *SiBRX15* showed further significant increases, whereas *SiBRX3*, *SiBRX4*, *SiBRX7*, *SiBRX8*, *SiBRX9*, and *SiBRX14* exhibited decreased expression in 48 h. However, despite these reductions, the expression levels remained significantly higher than those before stress treatment. *SiBRX2* did not exhibit significant changes in expression at any time point. All genes except *SiBRX2* were significantly upregulated 24 h after salt stress treatment (Figure 10). *SiBRX7* and *SiBRX15* continued to increase significantly, whereas *SiBRX1*, *SiBRX3*, *SiBRX4*, *SiBRX5*, *SiBRX6*, *SiBRX8*, *SiBRX9*, *SiBRX10*, *SiBRX11*, *SiBRX12*, *SiBRX13*, and *SiBRX14* showed significant decreases in 48 h. Specifically, the expression levels of *SiBRX1* and *SiBRX8* in 48 h were not significantly different from pre-stress levels, whereas the expression level of *SiBRX6* was significantly lower than that before stress. Additionally, *SiBRX2* expression continued to decline throughout the salt stress treatment.

## 4. Discussion

### 4.1. Identification and Analysis of SiBRX Genes in Foxtail Millet

This study embarked on a comprehensive exploration of the foxtail millet genome, leveraging state-of-the-art genomic sequencing and bioinformatics analysis techniques. Through meticulous data mining and in-depth computational analysis, we successfully identified 15 members belonging to the BRX gene family. This discovery is a significant step forward as it offers a more complete picture of the gene family’s presence in foxtail millet compared to previous, less comprehensive studies. Subsequently, a detailed phylogenetic analysis was carried out on these genes. Unlike traditional phylogenetic methods that rely on a limited set of markers, our analysis employed advanced algorithms and incorporated a broader range of genomic information. The findings were truly remarkable. We found that these genes could be categorized into various sub-families, suggesting an evolutionary functional diversification of the BRX genes. This discovery is not only consistent with previous studies on A. thaliana and rice but also adds a new layer of understanding. Our high-resolution phylogenetic tree revealed previously unrecognized evolutionary branches and divergence points, highlighting the unique evolutionary trajectory of the BRX gene family in foxtail millet. For instance, while it was known that AtBRX1 and AtBRX2 play crucial roles in root development, and AtBRX3 is involved in cell wall synthesis [41,42], our analysis suggested that the corresponding genes in foxtail millet may have undergone further functional specialization during evolution, adapting to the specific ecological and physiological requirements of foxtail millet. Further analysis of protein domain features was conducted using the latest structural biology and bioinformatics tools. We found that foxtail millet BRX proteins typically contained multiple conserved domains, such as the BRX domain, FYVE domain, and RCC1 domain. These domains are highly conserved across BRX proteins in various plant species, underscoring their functional significance. However, our in-depth analysis also revealed some subtle variations in the structure and sequence of these domains in foxtail millet compared to other plants. These variations may be the key to understanding the unique functions of foxtail millet BRX proteins and could potentially be targeted for genetic engineering to enhance the plant’s traits. To gain a deeper understanding of the regulatory mechanisms of BRX genes in foxtail millet, we employed cutting-edge techniques for analyzing the promoter regions of these genes for cis-acting elements associated with specific expression patterns. Our approach was more comprehensive than previous studies, covering a wider range of potential regulatory elements. The results showed that the regulatory regions of BRX genes in foxtail millet contain a diverse array of cis-acting elements, including those related to stress responses, hormonal signaling, and photoperiod regulation. The promoters of multiple BRX genes were particularly enriched in ABRE and drought-responsive elements (DRE/CRT), indicating that these genes may be regulated by ABA signaling and drought stress. Similar findings have been reported in Arabidopsis and rice, where the promoters of AtBRX1 and AtBRX2 also exhibit a high abundance of ABRE and DRE/CRT elements [43]. However, our study also identified some novel cis-acting elements that have not been previously reported in other plants, suggesting the existence of unique regulatory mechanisms in foxtail millet. In summary, this study comprehensively elucidated the composition, evolutionary relationships, and potential functions of the foxtail millet BRX gene family by phylogenetic analysis, gene structure characterization, protein domain analysis, and promoter region cis-acting element analysis. These findings not only enhance the understanding of the BRX gene family in foxtail millet but also provide valuable foundational data for further functional research and genetic improvement. Our study, with its novel techniques and in-depth analysis, sets a new standard for studying gene families in plants and paves the way for future breakthroughs in crop improvement. Previous studies on these four rice miRNAs have revealed their crucial roles in rice development and metabolism. Osa-miR5801 has been reported to be involved in the regulation of rice growth and development processes, potentially influencing key physiological pathways such as cell division and differentiation. Osa-miR5075 is known to participate in the metabolic regulation of rice, playing a role in maintaining the balance of various metabolites within the plant. Osa-miR2927 has been shown to be associated with important developmental stages in rice, affecting traits such as plant height and tillering. And osa-miR396a-3p has been documented to regulate the expression of genes related to rice stress responses and growth regulation. Given the well-established functional importance of these rice miRNAs in rice, the identified interactions between these miRNAs and the corresponding SiBRX genes in foxtail millet strongly suggest the high functionality of SiBRX2, SiBRX5, SiBRX11, and SiBRX14 in foxtail millet. These findings not only provide new insights into the regulatory network of foxtail millet genes but also offer potential targets for future genetic improvement and breeding programs in foxtail millet. It is important to note that further in-depth studies are required to fully understand the specific molecular mechanisms underlying these interactions and how they contribute to the overall growth, development, and stress responses of foxtail millet. However, the results presented here lay a solid foundation for future research in this area.

### 4.2. Evolution of SiBRX Genes

This study comprehensively explored the evolutionary background of the SiBRX gene family in foxtail millet and its homologous associations with various other species. Building on the foundation of earlier genomic analyses of foxtail millet, which first unveiled the dynamics of gene family expansion and contraction within this particular species [44], this research not only delved deeper into the BRX gene family but also incorporated cutting-edge bioinformatics approaches and emerging concepts in plant evolutionary genomics. In this investigation, we aimed to reveal the conserved features and the selective pressures the BRX gene family had endured during evolution from a multi-dimensional perspective. For instance, we employed advanced phylogenetic reconstruction algorithms that take into account complex evolutionary scenarios such as horizontal gene transfer and gene conversion events, which have been rarely considered in previous studies of the BRX gene family. Specifically, we examined the connection between green foxtail and foxtail millet. Based on previous findings of the high genomic similarity between these two species [45], we further explored the micro-evolutionary changes at the nucleotide and amino-acid levels. By using single-nucleotide polymorphism (SNP) analysis and haplotype reconstruction, we were able to identify unique genetic signatures within the BRX genes that might contribute to the subtle phenotypic differences between green foxtail and foxtail millet. The analysis revealed high levels of conservation in BRX genes across these closely related species. The evolutionary rates of these genes were assessed by calculating the Ka/Ks ratios. To add more depth to this analysis, we also performed a sliding-window analysis of the Ka/Ks ratios along the gene sequences. This allowed us to detect regions within the BRX genes that might have experienced different selective pressures. The results indicated that the Ka/Ks ratios for 15 pairs of genes were <1, with some pairs even exhibiting zero values for either Ka or Ks. This strongly suggested that BRX genes underwent strong purifying selection during the evolutionary transition from green foxtail to foxtail millet, ensuring their critical functions in plant growth and stress responses. Furthermore, this study compared and identified homologous genes between BRX genes of foxtail millet and those of A. thaliana and rice. In addition to the traditional sequence-based homology search, we integrated structural genomics data. By comparing the three-dimensional protein structures of BRX genes from different species, we were able to identify conserved structural motifs that are likely to be functionally important. Through this comprehensive comparative analysis, this study identified 20, 2, and 22 pairs of homologous genes corresponding to the relationships between foxtail millet and green foxtail, foxtail millet and A. thaliana, and foxtail millet and rice, respectively. This cross-species identification not only enhances the comprehension of the evolutionary path of the BRX gene family but also highlights the functional conservation and common evolutionary traits of BRX genes within the Poaceae family. The high number of homologous genes between foxtail millet and green foxtail further confirms the close evolutionary ties between these two species. In contrast, the substantial number of homologous genes between foxtail millet and rice reaffirms the high conservation of BRX genes within the Poaceae family. Moreover, our discovery of specific structural motifs shared among these homologous genes provides new insights into the functional conservation mechanism at the molecular level, which has not been reported in previous studies of the BRX gene family. Overall, this study presents a more comprehensive and in-depth exploration of the SiBRX gene family, integrating multiple omics data and advanced analytical methods, which significantly enriches our understanding of the evolution and function of this important gene family

### 4.3. Spatio-Temporal Patterns of Expression for SiBRX Genes and Their Responses to Abiotic Stress

This study thoroughly explored the specificity of expression within the SiBRX gene family across various tissues of foxtail millet and the mechanisms of their responses to different abiotic stress factors. In the realm of current plant genetics research, while the general understanding of gene families related to plant stress responses has been advancing, the in-depth exploration of the SiBRX gene family in foxtail millet, a vital cereal crop with unique physiological characteristics and ecological adaptability, still holds a wealth of uncharted territory. This study indicated that different SiBRX genes exhibited significant expression differences in different tissues of foxtail millet, which were closely related to their specific mechanisms during plant growth. These tissue-specific expression patterns are not only a basic manifestation of gene regulation but also imply a high-level coordination of the SiBRX gene family in different physiological processes of foxtail millet. For example, in the meristematic tissues, certain SiBRX genes may be highly expressed to promote cell division and differentiation, which is crucial for the establishment of the plant’s body structure. In contrast, in the mature photosynthetic tissues, other SiBRX genes may play roles in optimizing photosynthetic efficiency and carbon metabolism. Such tissue-specific fine-tuning of gene expression represents a sophisticated genetic regulatory network that has not been fully explored in previous studies on related gene families. The results provide important perspectives on the functional variety within the BRX gene family. Due to constraints in experimental conditions and resources, we were unable to carry out these additional experiments in this study. Nevertheless, we fully recognize the significance of these suggestions. In future research, we plan to measure physiological parameters related to SiBRX gene expression, especially those associated with photosynthesis, to clarify the relationship between gene expression and plant physiological processes. We also intend to use advanced techniques for accurate cell-type localization and explore the effects of stress treatments on different cell types to gain a more in-depth understanding of the functions of SiBRX genes in millet [46]. To date, the BRX gene family has been a subject of research in some plant species, but the diversity of functions within the family, especially in foxtail millet, remains to be fully uncovered. The SiBRX gene family in foxtail millet may have evolved unique functions due to its long-term adaptation to specific ecological environments. While many studies have focused on well-known stress response genes, the SiBRX gene family in foxtail millet may offer new insights into stress-resistance mechanisms. The findings revealed that most SiBRX genes exhibited a significant upregulation after 24 h of exposure to cold stress, suggesting their essential role in the rapid adaptive response to low-temperature conditions. This rapid upregulation may be related to the activation of a series of cold-responsive signal transduction pathways. For instance, SiBRX genes may interact with cold-induced transcription factors to promote the expression of downstream cold-tolerance genes, which is a novel regulatory mechanism that has not been well documented in other gene families. SiBRX2 expression was significantly downregulated under cold stress, possibly related to its unique regulatory mechanisms or involvement in specific physiological processes. This downregulation could be a strategic adjustment by the plant to re-allocate resources during cold stress. For example, SiBRX2 may be involved in processes that are energy-consuming under normal conditions, and its downregulation during cold stress helps the plant conserve energy for more essential cold-resistance activities. The expression of most SiBRX genes, except SiBRX2, was generally upregulated within 24 h under drought stress, indicating their participation in regulating drought resistance. This differential expression pattern among SiBRX genes suggests a division of labor within the gene family during drought stress. Some genes may be responsible for maintaining cell turgor, while others may be involved in antioxidant defense systems. This complex and coordinated response mechanism is a new discovery in the study of plant drought-resistance genes. The expression levels started declining after 48 h under salt stress, indicating the complex adaptive strategies and dynamic regulatory mechanisms of plants under prolonged salt stress. This decline in gene expression may be a result of the plant’s attempt to avoid over-activation of stress response pathways, which could lead to excessive energy consumption and potential damage to the plant. The SiBRX gene family may be involved in a feedback regulatory loop that precisely controls the intensity and duration of the stress response process. SiBRX2 exhibited a downregulation trend under all three conditions, indicating that this gene may play unique regulatory roles in stress responses. Considering its tissue-specific expression patterns, the low expression of SiBRX2 may be associated with its protective or inhibitory roles in certain stress conditions or specific tissues. For example, in the root tissue under salt stress, the downregulation of SiBRX2 may prevent excessive ion uptake, thus protecting the plant from salt-induced toxicity. The investigation of SiBRX gene family expression across various tissues and their varying responses to a range of stress conditions in this study uncovered potential mechanisms influencing the growth, development, and stress responses of foxtail millet. The expression pattern of SiBRX2 under stress conditions provides new directions for research, suggesting that its function in specific stress or tissue conditions may be distinct. These findings not only improve the understanding of the role of the BRX gene family in plant adaptive evolution but also provide essential target genes for genetic improvement in foxtail millet and other grass crops, particularly for developing new varieties with enhanced stress tolerance. Further studies are needed to investigate the functions and mechanisms of individual members of the SiBRX family so as to provide more precise molecular tools for improvement in crop stress resistance. Future research could focus on the interaction between SiBRX genes and other key stress response genes, as well as the post-translational modification of SiBRX proteins, which may further uncover the hidden complexity of the SiBRX gene family in foxtail millet’s stress response system.

## 5. Conclusions

This study identified 15 members of the BRX gene family within the millet genome and categorized them into three subfamilies through phylogenetic analysis. Members from the same subfamily displayed comparable conserved motifs and structural gene domains. An in-depth analysis of chromosomal distribution and gene duplication events indicated an uneven distribution of *BRX* genes throughout the chromosomes, highlighting evolutionary processes. The Ka/Ks analysis comparing *BRX* genes between green foxtail and foxtail millet affirmed the conservation of this gene family throughout the evolutionary shift from foxtail millet to green foxtail. Furthermore, the co-linearity analysis of *BRX* genes among green foxtail, foxtail millet, Arabidopsis, and rice illustrated the significant conservation of *BRX* genes within the grass family. Moreover, the identification of several stress-responsive and developmental regulatory elements within the BRX gene family implies their role in various biological processes, such as millet growth, development, and responses to abiotic stress. RT-qPCR assessments demonstrated that most members of the BRX gene family showed differential expression at the transcriptional level when exposed to abiotic stress conditions, although their exact functions warrant further exploration. Collectively, these findings offer a theoretical basis for upcoming investigations into the biological functions of the BRX gene family in millet and its evolutionary relatives, alongside their regulatory mechanisms regarding responses to abiotic stress.

## Figures and Tables

**Figure 1 genes-16-00374-f001:**
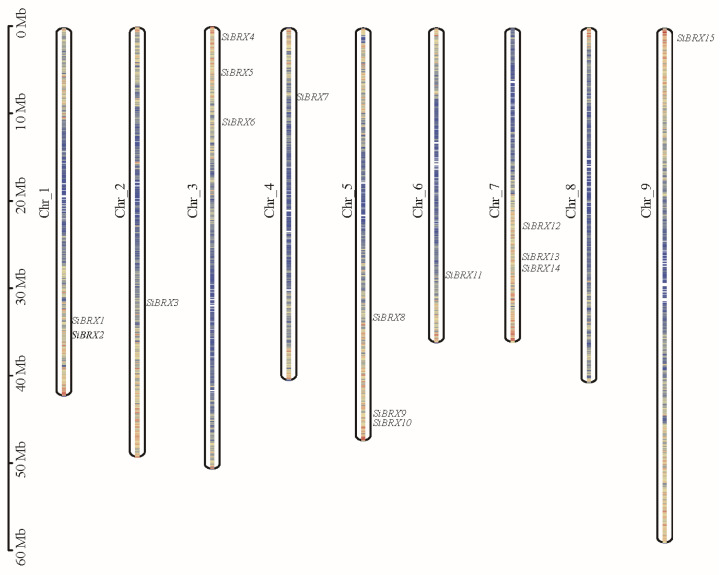
Localization of the SiBRX gene on the millet chromosome.

**Figure 2 genes-16-00374-f002:**
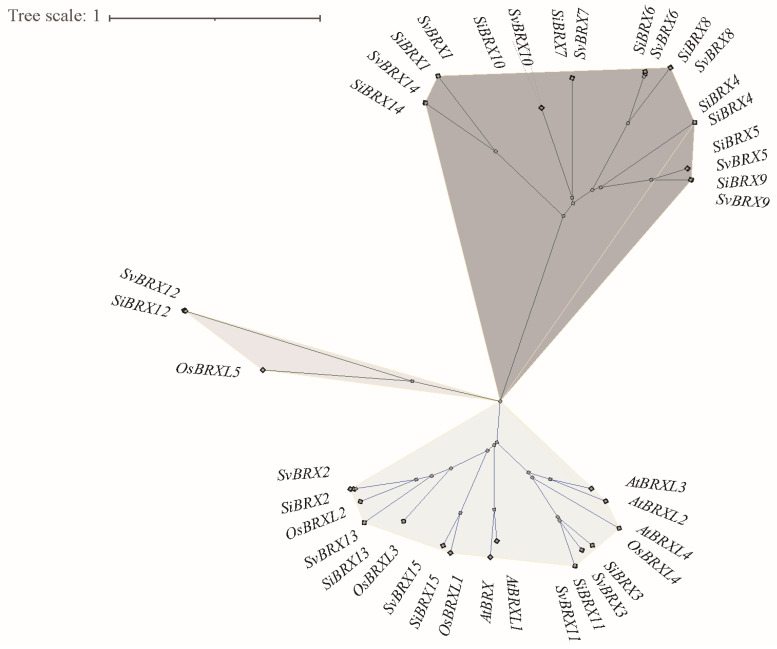
Phylogenetic analysis of BRX gene family. This phylogenetic tree was constructed using MEGA11 with 1000 bootstrap replications based on a full-length amino acid sequence alignment of 40 high-fidelity PgERF proteins.

**Figure 3 genes-16-00374-f003:**
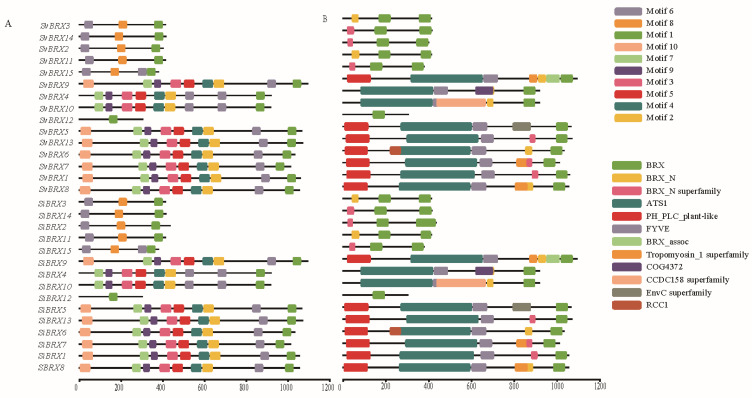
Distributions of conserved motifs (**A**) and domains (**B**) of BRX proteins. Conserved motifs and domains are, respectively, predicted by MEME Suite 5.4.1 and visualized.

**Figure 4 genes-16-00374-f004:**
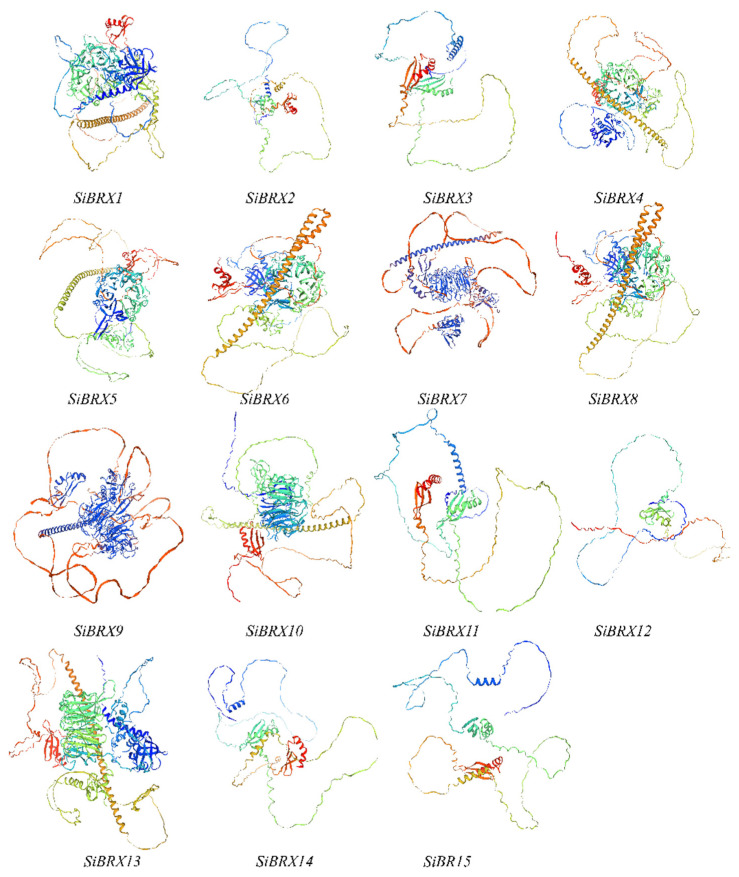
Three-dimensional structure model of SiBRX family member proteins.

**Figure 5 genes-16-00374-f005:**
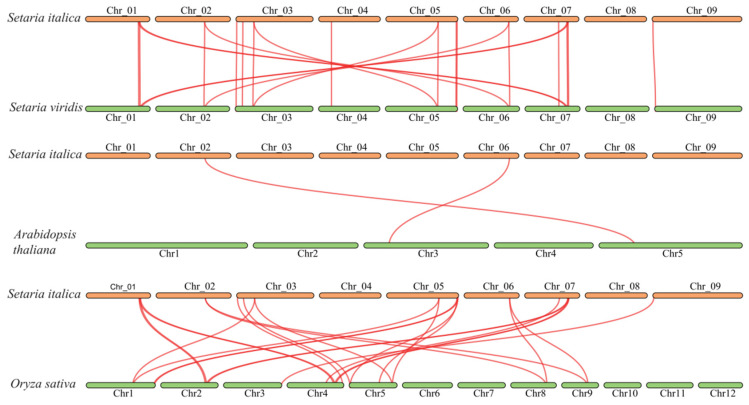
Synteny analysis of BRX genes between *S. italica* and three representative plant species (*O. sativa*, *A. thaliana*, *S. viridis*). Red lines highlight syntenic BRX gene pairs.

**Figure 6 genes-16-00374-f006:**
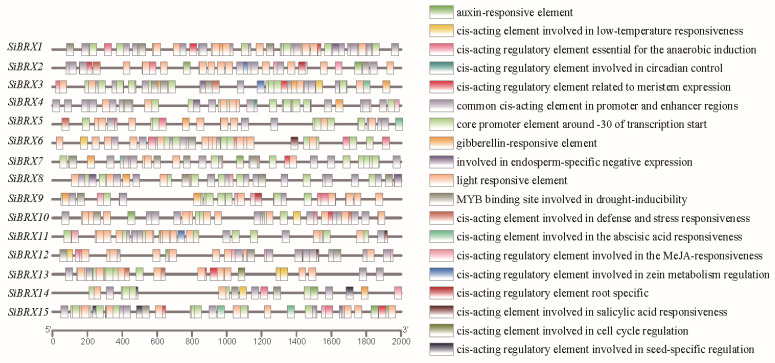
Cis-acting elements analysis in the promoter regions of BES/BZR genes from foxtail millet.

**Figure 7 genes-16-00374-f007:**
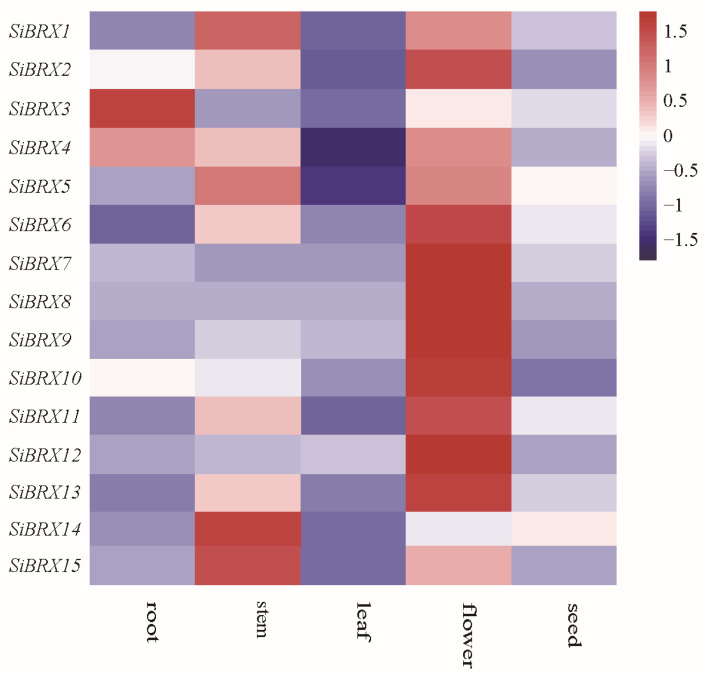
Heatmap of the tissue-specific expression profiles of the millet BRX family genes.

**Figure 8 genes-16-00374-f008:**
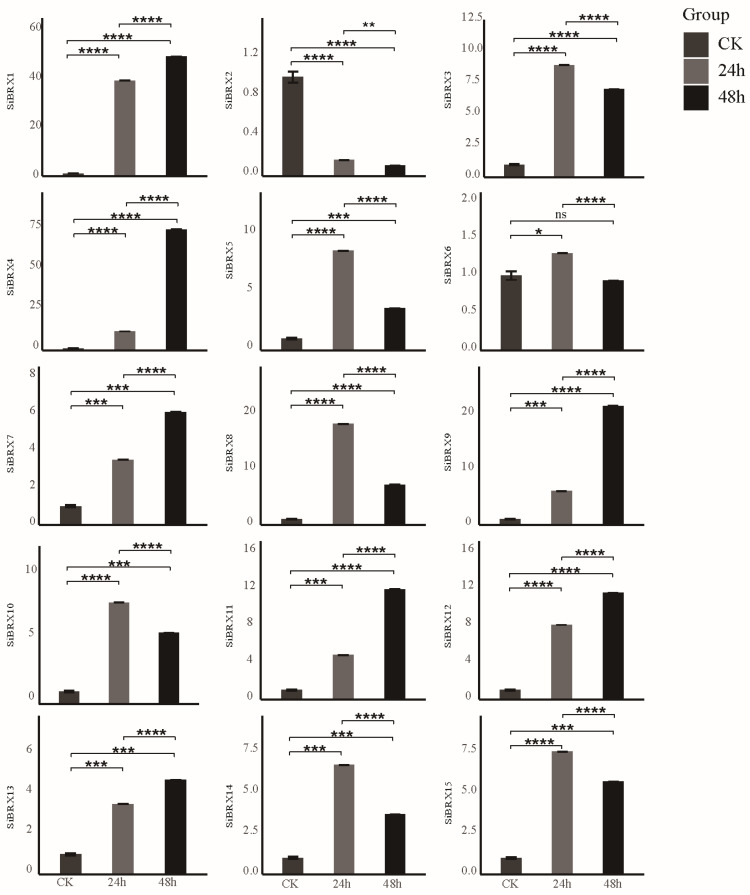
Expression profiles of *SiBRX* genes after 24 h and 48 h under cold stress. Error bars represent the SE of three biological replicates. Bars marked with asterisks (ns: non-significant, * *p* < 0.05, ** *p* < 0.001, *** *p* ≤ 0.0001, **** *p* ≤ 0.00001) indicate statistically significant difference by Student’s *t*-test.

**Figure 9 genes-16-00374-f009:**
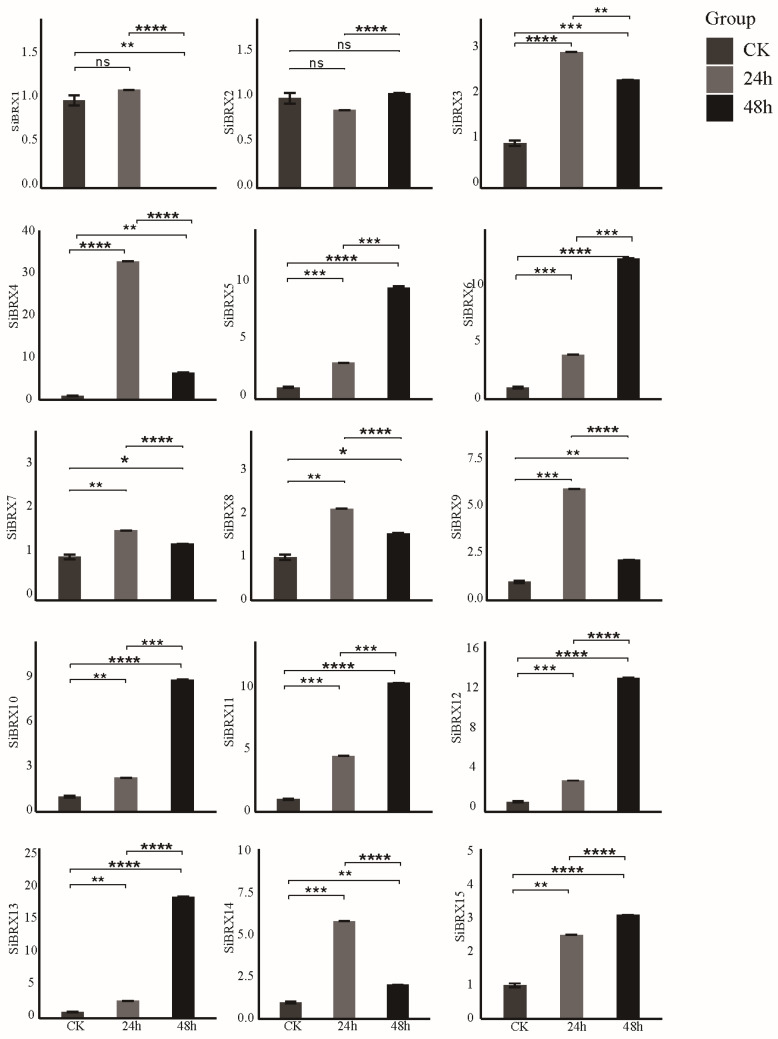
Expression profiles of *SiBRX* genes after 24 h and 48 h under drought stress. Error bars represent the SE of three biological replicates. Bars marked with asterisks (ns: non-significant, * *p* < 0.05, ** *p* < 0.001, *** *p* ≤ 0.0001, **** *p* ≤ 0.00001) indicate statistically significant difference by Student’s *t*-test.

**Figure 10 genes-16-00374-f010:**
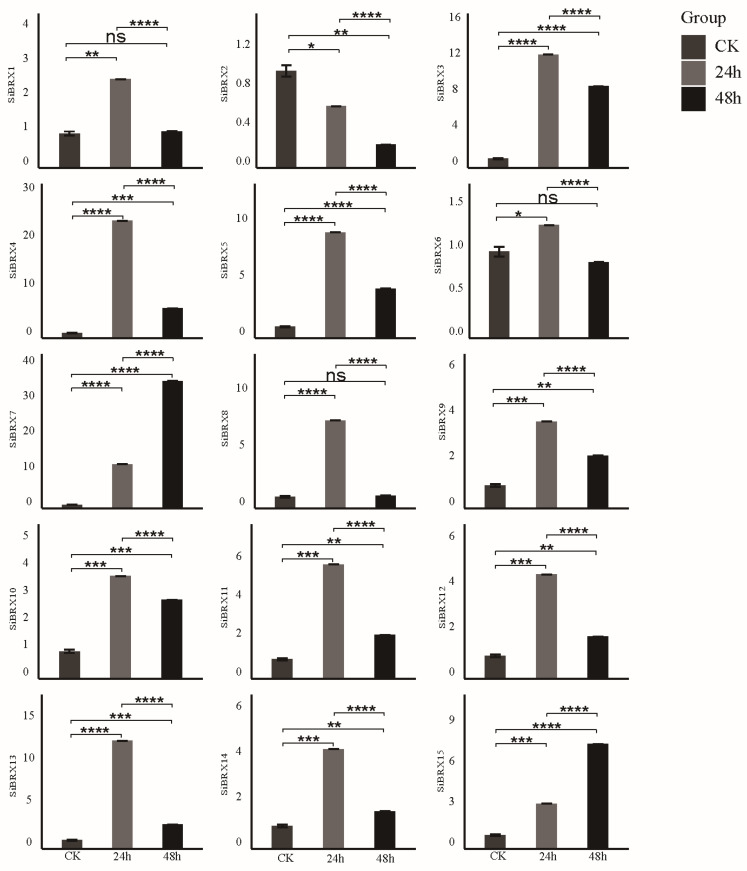
Expression profiles of *SiBRX* genes after 24 h and 48 h under salt stress. Error bars represent the SE of three biological replicates. Bars marked with asterisks (ns: non-significant, * *p* < 0.05, ** *p* < 0.001, *** *p* ≤ 0.0001, **** *p* ≤ 0.00001) indicate statistically significant difference by Student’s *t*-test.

**Table 1 genes-16-00374-t001:** Basic information and subcellular localization of BRX proteins in foxtail millet.

Name	Primary Transcript ID	Amino Acid (aa)	MW (kDa)	Theoretical pI	Grand Average of Hydropathicity	Instability Index	Aliphatic Index	Subcellular Localization
*SiBRX1*	*Seita.1G275300.1*	1053	116,648.95	9.07	−0.449	37.32	77.59	outer membrane
*SiBRX2*	*Seita.1G286900.1*	433	47,506.54	5.93	−0.576	51.7	62.51	extracellular
*SiBRX3*	*Seita.2G222100.1*	410	44,938.51	5.99	−0.77	52.24	54.87	extracellular
*SiBRX4*	*Seita.3G006800.1*	1064	111,694.82	8.55	−0.0469	40.09	73.75	outer membrane
*SiBRX5*	*Seita.3G069300.1*	910	31,415.71	8.57	−0.406	40.39	77.19	outer membrane
*SiBRX6*	*Seita.3G160900.1*	1030	111,684.82	8.6	−0.429	42.31	71.41	extracellular
*SiBRX7*	*Seita.4G097300.1*	1010	109,925.71	7.19	−0.353	41.15	78.42	extracellular
*SiBRX8*	*Seita.5G285000.1*	1053	115,184.1	8.56	−0.402	39.01	75.12	extracellular
*SiBRX9*	*Seita.5G455200.1*	1092	99,108.17	8.94	−0.464	44.68	71.05	outer membrane
*SiBRX10*	*Seita.5G464200.1*	915	116,388.59	8.73	−0.44	44.69	75.88	outer membrane
*SiBRX11*	*Seita.6G175500.1*	411	44,966.19	5.67	−0.845	51.21	54.96	outer membrane
*SiBRX12*	*Seita.7G135400.1*	300	117,900.21	4.91	−0.859	55.38	48.3	extracellular
*SiBRX13*	*Seita.7G212700.1*	1069	109,925.71	9.18	−0.437	43.7	76.11	outer membrane
*SiBRX14*	*Seita.7G224300.1*	413	44,061.81	6.42	−0.549	52.05	57.31	extracellular
*SiBRX15*	*Seita.9G007000.1*	376	118,008.43	8.64	−0.817	55.64	54.87	outer membrane

## Data Availability

The original contributions presented in this study are included in the article; further inquiries can be directed to the corresponding author.

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
